# Evaluation of Spent Grain Biochar Impact on Hop (*Humulus lupulus* L.) Growth by Multivariate Image Analysis

**Tiziana Amoriello** [1],* , **Simona Fiorentino** [2], **Valerio Vecchiarelli** [2] **and Mauro Pagano** [3],*

1   CREA Research Centre for Food and Nutrition, Via Ardeatina 546, 00178 Roma, Italy
2   Centro Appenninico del Terminillo Carlo Jucci, Università degli Studi di Perugia, Via Comunali 43, 02100 Rieti, Italy; simona.fiorentino@unipg.it (S.F.); valerio.vecchiarelli@unipg.it (V.V.)
3   CREA Research Centre for Engineering and Agro-Food Processing, Via della Pascolare 16, 00015 Monterotondo (Roma), Italy
*   Correspondence: tiziana.amoriello@crea.gov.it (T.A.); mauro.pagano@crea.gov.it (M.P.); Tel.: +39-06-514941 (T.A.)

**Abstract:** Biochar is generally considered as an effective soil amendment, which can improve soil organic matter and nutrients content and enhance crop productivity. In this study, biochar derived from brewers' spent grain (BSG) was used in a pot and field experiment to assess whether its addition to soil could affect hop plant growth. The experiment was conducted in Central Italy during the period March–August 2017. Three different German cultivars of hop plant (Hallertau Magnum, Perle, Spalter spalt) were considered. Biochar was added to the pot soil at 20% level. Its effect on the roots was evaluated using multivariate image analysis (MIA) and the statistical technique of general linear models (GLM), whereas the shoots, bines length and yield using GLM. Results showed that biochar significantly improved root growth ($p < 0.0001$). Regarding shoots, no variability for the genotypes was observed during the vegetative period, whereas slight differences resulted before plant dormancy, especially for the Hallertau Magnum cultivar. No differences in the number of leaves or bines length were observed between the two treatments for all cultivars. The addition of biochar to the soil significantly improved yield (number of cones). These results highlighted that BSG-derived biochar can be useful to improve hop plant growth and cones production.

**Keywords:** amendment; biochar; brewers' spent grain; hop; image analysis; plant growth

## 1. Introduction

Beer is one of most consumed beverages in the world, and plays an important role in the global economy. In 2018, the overall beer production amounted to about 1.94 billion hectolitres. However, beer processing produces a huge amount of waste: for every hectolitre of beer produced, about 20 kg of spent grain, 0.2–0.4 kg of spent yeast and 0.3 kg of spent hops/hot trub are generated. In particular, the 85% of total solid waste is represented by brewers' spent grain (BSG), the residue left after barley malting and separation of the wort during the brewing process, and it contains the husk and the outer layer of barley kernel [1]. BSG is mainly composed of protein (more than 20%) and fibers, represented by a lignino-cellulosic material, whose main constituents are hemicellulose (28–35%), cellulose (17–25%) and lignin (7–27%) [1–3].

Appropriate management of these waste streams has become a challenging issue. The solid by-products from the brewing process are mainly disposed of as waste or sold as animal feed, due to their considerable amount of valuable compounds (proteins, lipids, carbohydrates, polyphenols, and minerals) and nitrogen-containing nutrients. However, brewing industries are interested in new

solutions to reduce the amount of waste produced and transform by-products in added value products. Therefore, in a perspective of a zero waste approach and in order to reduce waste storage and logistics costs, alternatives to valorize these residues and to recover and reuse them in a sustainable and profitable way are continuously proposed and developed. Moreover, by-products can be transformed in combustible gas, to be reused in the beer production cycle, or in soil amendments for agricultural applications [4,5].

A valid alternative can be the transformation of BSG into biochar, which can be used as an effective soil amendment for the production of vegetables. Biochar is a carbon-rich, fine-grained and porous material with stable physical and chemical properties, produced from waste biomass through pyrolysis, e.g., a thermal decomposition of lignocellulose biomass by heating at elevated temperatures (generally between 350 and 700 °C) under limited oxygen conditions [6–8]. It has stable aromatic C structures, low O and H to C ratios, low bulk density, moderate cation exchange capacity (CEC), and high ash content, pH and surface area [9,10]. Due to these properties, it is recognized as a multifunctional material which can be applied for long-term C sequestration and climate change mitigation [11,12]. It can be used successfully in agriculture to increase the organic C content of soil and reduce the leaching of nutrients; it could have positive effects on soils contaminated with heavy metals and organic pollutants [6,12–15]. Furthermore, biochar can be useful as soil amendment, due to its high porosity and sorption capacity and large surface area, reducing soil bulk density, improving soil structure, and increasing soil water holding capacity [7]. Biochar application can increase plant root growth, root penetration, and nutrient and water uptake [16]. In fact, N concentration in soil is increased by biochar during critical stages of plant growth, and N uptake and fertilizer recovery from roots are mostly guaranteed [7,17]. Biochar amendment also significantly enhances microbial activity and abundance, probably due to a greater adsorption of various nutrients [18,19]. Such positive influence could have effects on soil structure and indirectly on plant growth and rooting patterns [20].

Numerous studies have been conducted in controlled environments to evaluate the impact of biochar on soil properties and plant growth. However, the effects are not always positive: benefits on soil and plant growth strongly depend on the feedstock sources used to produce the biochar, production conditions (pyrolysis temperature, heating rate and residence time), differences in soil properties and specie-specific root growth patterns [15,21–23]. Therefore, this study aims to assess the potential ability of BSG-derived biochar for soil amendment and its influence on hop plant growth.

## 2. Materials and Methods

### 2.1. Biochar Production and Characterization

A fifty kilograms sample of wet BSG was chosen randomly from a big container at a craft brewery. Subsequently, three replicates of about 1 kg each were selected randomly from this sample, oven-dried at 105 ± 1 °C for 78 h and successively subjected to elemental analysis for the determination of the CHN content, according to UNI EN 15104: 2011. Ash content was analyzed by heating ground dried BSG samples (maximum grain size 2 mm) at 550 °C ± 1 °C for 8 h in a muffle furnace, according to UNI EN 14775: 2010. Three analytical replicates were performed for all parameters analyzed and for each sample.

Biochar was prepared from BSG using the pyrolytic reactor Elsa D17 (pat. BLUECOMB) and through a thermochemical process that can reach carbonization temperatures of 400–500 °C [24,25].

Biochar was chemically characterized by Fourier transformed infrared spectroscopy analysis (FT-IR), according to the methodology described by Amoriello et al. [26]. Spectra were collected at room temperature with a FT-IR spectrometer (iS 50 FT-IR Nicolet Thermo Fisher Scientific Inc., Waltham, MA, USA) equipped with a single-reflection horizontal ATR cell with a diamond crystal [26].

### 2.2. Botanical and Agronomic Characteristics of Hop Plant

Hop (*Humulus lupulus* L.) is a dioecious perennial plant, belonging to the *Cannabaceae* family. It reaches maturity after the first three years and remains productive for over 20–25 years [27]. The hop plant consists of a perennial rootstock ("crown") of rhizomes below ground, annual climbing bines above ground, which provide the canopy and photosynthetic capacity to support flowering, and flowers that develop at the terminal buds of lateral branches and are harvested as green cones. The perennial root system can grow more than 4 m deep and up to 5 m laterally. Rhizomes feed the growth of the productive canopy and ensure the survival of the plant from successive seasons. The growing season spans from March to August–September in the northern hemisphere. Hops emerge in early spring and grow up to a height of 5–7 m on poles or under a trellis system. Flowering starts in late June or early July in the northern hemisphere. The cones mature for picking between August and September, depending on climate conditions and genotype. The female strobiles (cones) represent the most interesting parts of the plant from a technological point of view, as they are one of the essential ingredients of the brewing industry, providing aroma, bitterness, flavor, and antimicrobial properties to beer [28].

### 2.3. Experimental Site and Design

The study was carried out in Rieti, Italy (latitude 42°24′29″52 N, longitude 12°51′36″36 E) during the period March–August 2017. The sampling site is characterized by relatively cold and rainy winters and hot and dry summers. The long-term average annual temperature recorded between 1980 and 2009 was 12.1 °C; the mean temperature during the growing season was 16.0 °C; the average maximum temperature for the summer months was 30.1 °C; the average minimum temperature for the coldest months was −1.6 °C. In summer, maximum temperatures were often over 30 °C. The annual mean precipitation was equal to 1021 mm and it fell mainly from October to December. Summer rainfall was irregularly distributed and the total mean amount over June–August was 136 mm. In 2017, the annual average temperature was 12.7 °C; the average temperature in the growing season was 17.4 °C; the average maximum temperature for the hottest months (July to August) was 33.2 °C. The annual precipitation, precipitation for the growing season, and precipitation over the hottest months was 721 mm, 186 mm, and 27 mm, respectively. The annual relative humidity was 68%.

A completely randomized experimental design was conducted during the hop growing season in a climate uncontrolled greenhouse environment from March to 5 May and outside from 6 May to 28 August. Three different cultivars of German hop plant (Hallertau Magnum, Perle, Spalter spalt) were considered for the study due to their economic importance and their sensitivity to environmental conditions [27]. These three varieties grow well in most climates, but best in warm, dry and sunny regions. Hallertau Magnum produces very good yields (1700–2300 kg ha$^{-1}$); the cone's structure is very large and longish size. Perle also produces good yields (1600–2100 kg ha$^{-1}$); cone size is small to medium. Spalter spalt yield amounts 1750–2000 kg ha$^{-1}$; the cone size is small to medium.

In March, hop rhizomes were placed in plastic pots (15 cm × 14 cm) with soil containing acid peat, expanded perlite and clay. Each pot contained 8 L of soil. The packed pot soil had a pH of 6.0, a dry bulk density of 1.1 g cm$^{-3}$, an electric conductivity of 0.45 dS m$^{-1}$ and a porosity of 90% *v/v*. Hop rhizomes for each variety were randomly assigned between pots with soil added with 20% of biochar (1.6 L of biochar and 6.4 L of soil containing acid peat, expanded perlite and clay) and pots without biochar (control). Trials with biochar were replicated seven times, whereas controls (without biochar) were run in triplicate. We chose an unbalanced sampling to better evaluate the variability of the biochar effect on hop plants. Young plants were transferred from pots to the field at the beginning of May, corresponding to the moment of bines elongation. No biochar was added in field. Until June, the plants were irrigated with 0.5 L of water every two days; in the two warmest months (July and August), the same amount of water was given every day.

Plant development was monitored through the percentage of roots in pot (as described in the Section 2.4) and the number of leaves at the beginning of May; shoot diameter was measured by caliper

at five different times (T1: 30 June, T2: 18 July, T3: 27 July, T4: 3 November, T5: 21 November), i.e., before flowering, near cones maturity, before plant dormancy; measurement of climbing bines length and number of cones at maturity.

*2.4. Statistical Analysis and Multivariate Image Analysis*

In order to quantify the effect of biochar on root growth for each genotype, a non-destructive technique, e.g., a multivariate image analysis, was carried out according to Fongaro et al. [29]. At the beginning of May, plants were extracted from pots and the images of roots and soil for each plant were acquired twice using a digital camera Nikon D750 at a high resolution and a color depth of 16 bits, saving the captured images in uncompressed RAW format. The images were acquired by photographing the basis and four external opposite sides. To create the final data set, a region of interest (ROI) of $472 \times 472$ pixels, representative of the whole sample surface, was extracted from each image using the Lightroom Classic CC 2018.

The images were processed by PLS Toolbox 8.5 (Eigenvectors Research, Inc., Manson, WA, USA) for MATLAB R2016b (The MathWorks Inc., Natick, MA, USA). Red (R), green (G), blue (B) values, measures in the RGB space color, were calculated. A principal component analysis (PCA) on the RGB values was carried out, providing the pixel distribution in the score space (score plot). The first principal component (PC1) contained most of the original information, with a decreasing amount in the remaining score images. PC1 score image. PC1 score image was converted from the RGB space color into the HSI (hue, saturation, intensity) space color. Then, the Hue channel was extracted to obtain the hue image. A pixel segmentation in different intensity ranges, corresponding on different material (soil and root), was carried out to select roots. In this way, it was possible to quantify the percentage of roots present in each pot and for each cultivar.

The influence of genotype, treatment and the two-way interaction on roots percentage or number of leaves or bines length or number of cones was tested using the technique of General Linear Models (GLM). The statistical model was:

$$Y_{ij} = \beta_0 + \beta_1 \cdot \tau_i + \beta_2 \cdot \delta_j + \beta_3\, \tau_i \cdot \delta_j + \varepsilon_{ij} \tag{1}$$

where $Y_{ij}$ = roots percentage or number of leaves or bines length or number of cones, $\beta_0$ = mean effect common to all observations; $\beta_{1,2,3}$ = unknown regression parameters; $\tau_i$ = treatment ($i$ = 1, 2; 1 = with biochar, 2 = without biochar); $\delta_j$ = cultivar ($j$ = 1, 2, 3; 1 = Hallertau Magnum, 2 = Perle, 3 = Spalter spalt); $\varepsilon_{ij}$ = error term.

GLM was also applied to monitor the plant development in time. The statistical model was:

$$Y_{ijk} = \beta_0 + \beta_1\, \tau_i + \beta_2\, \delta_j + \beta_3\, \gamma_k + \beta_4\, \tau_i \cdot \delta_j + \beta_5 \tau_i \cdot \gamma_k + \beta_6\, \delta_j \cdot \gamma_k + \beta_7\, \tau_i \cdot \delta_j \cdot \gamma_k + \varepsilon_{ijk} \tag{2}$$

where $Y_{ijk}$ = diameter of shoots, $\beta_0$ = mean effect common to all observations; $\beta_{1-7}$ = unknown regression parameters; $\tau_i$ = treatment ($i$ = 1, 2; 1 = with biochar, 2 = without biochar); $\delta_j$ = cultivar ($j$ = 1, 2, 3; 1 = Hallertau Magnum, 2 = Perle, 3 = Spalter spalt); $\gamma_k$ = time ($k$ = 1, … ,5; 1 = 30 June, 2 = 18 July, 3 = 27 July, 4 = 3 November, 5 = 21 November); $\varepsilon_{ijk}$ = error term.

Post hoc Tamhane test ($p < 0.05$) was carried out in order to evaluate differences between groups (with or without biochar) for each considered variable. Statistical analysis was performed with SPSS 20.0 software (SPSS, Inc., Chicago, IL, USA).

## 3. Results

*3.1. Brewers' Spent Grain and Biochar Characterization*

BSG analysis showed an average moisture content of 8.8%, and a content of ashes, C, H, N contents of 5.3%, 45.7%, 9.0% and 4.2%, respectively, on the dry weight basis.

Brewers' spent grain (BSG) and biochar were characterized by FT-IR spectra (Figure 1).

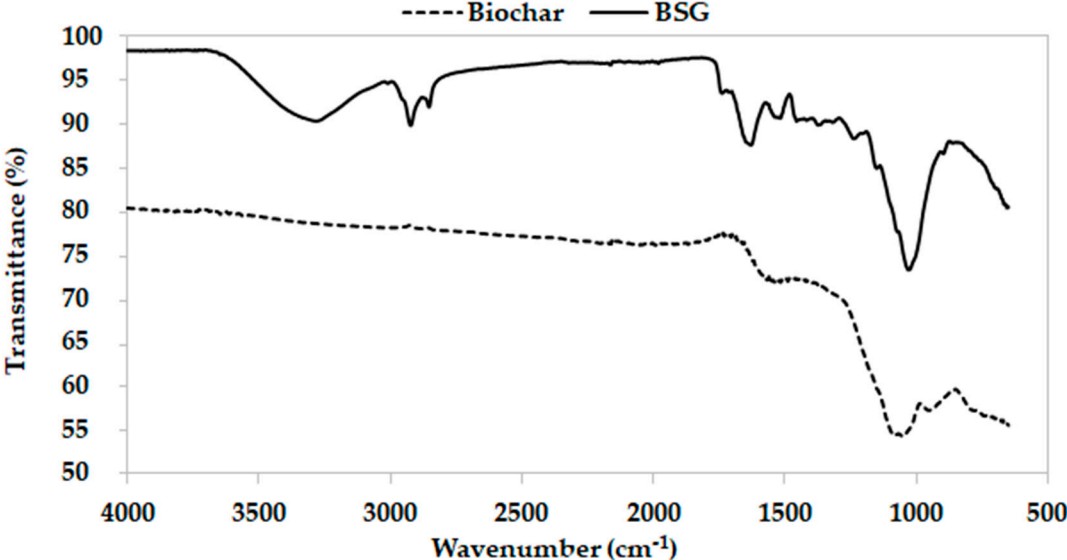

**Figure 1.** FT-IR spectra of brewers' spent grain (BSG) and biochar.

The two spectra showed different profiles and changes of functional groups, demonstrating the influence of the pyrolytic treatment on the chemical structure of the raw material. In fact, the organic matter pyrolysis caused water loss and variation in concentration of mineral components due to the heat-induced mass loss [30]. BSG profile showed a peak at 3274 cm$^{-1}$ corresponding to a vibrational band consistent with hydroxyl groups (O-H); two peaks at 2926 and 2853 cm$^{-1}$, corresponding to aliphatic C-H stretch; a peak at 1742 cm$^{-1}$, associated with the presence of aldehydes, ketones and carboxyl groups (C=O); a band with a peak at 1633 cm$^{-1}$, assigned to aromatic lignin components (C=C); a peak at 1030 cm$^{-1}$, corresponding to C-O-C stretch. The thermal decomposition strongly influenced the intensity of biochar bands from 4000 to 1600 cm$^{-1}$ (carboxylic bonds, amides and aliphatic hydrocarbon) [30]. In particular, the disappearance of the peaks at 2926 and 2853 cm$^{-1}$ may be due to the fact that methyl groups, which are the weakest functional groups, as well as the OH groups, break during pyrolysis. The CH peaks shift from being more aliphatic to more aromatic (and eventually disappear altogether) [31]. The cleavage of these groups contributes to mass loss during thermal decomposition and to the production of "non-condensable" gas. Moderate differences between BSG and biochar profiles from 1600 to 1500 cm$^{-1}$ may be attributed to the formation of carbonate-carboxyl group during pyrolysis. At last, biochar showed similar functional groups, between 600 and 1500 cm$^{-1}$. These results were in accordance with previous studies on BSG-derived biochar [32,33]. The FT-IR analysis highlighted the presence of functional groups such as hydroxyl and carboxyl groups, necessary to consider biochar a soil amendment for improving of the cation exchange capacity and as a potential adsorbent [34].

### 3.2. Multivariate Image Analysis

Multivariate image analysis was applied to extract roots region and to quantify differences in percentage for the two treatments, with and without biochar. Representative images from roots and soil samples for each cultivar and treatment after RGB processing and PCA score images are shown in Figure 2.

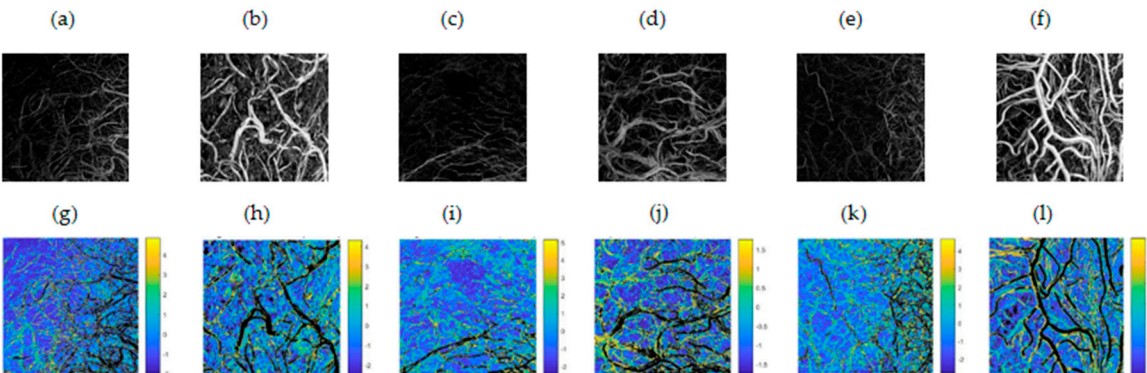

**Figure 2.** RGB and PCA score images of roots and soil samples representative of different cultivars and treatments, obtained from image analysis process. (**a**) RGB image of Hallertau Magnum without biochar; (**b**) RGB image of Hallertau Magnum with biochar; (**c**) RGB image of Perle without biochar; (**d**) RGB image of Perle with biochar; (**e**) RGB image of Spalter Spalt without biochar; (**f**) RGB image of Spalter Spalt with biochar; (**g**) PCA score image of Hallertau Magnum without biochar; (**h**) PCA score image of Hallertau Magnum with biochar; (**i**) PCA score image of Perle without biochar; (**j**) PCA score image of Perle with biochar; (**k**) PCA score image of Spalter Spalt without biochar; (**l**) PCA score image of Spalter Spalt with biochar.

For all the images, the PC1 score image explained all the variance, being related to pixels of both the root and soil. Comparing the PC1 score images and the relative RGB images, it was possible to distinguish areas with different color intensity, indicated with blue (soil) and yellow (root). Thus, the same false color was assigned to the pixels having the same characteristics, well distinguishing the two regions in the PC1 score images. The pixel segmentation allowed the extraction of the root areas (in black) and the quantification of the roots percentages. Results obtained were plotted in Figure 3. Box plots showed a strong influence on root growth present in the pots with biochar in comparison to those without biochar.

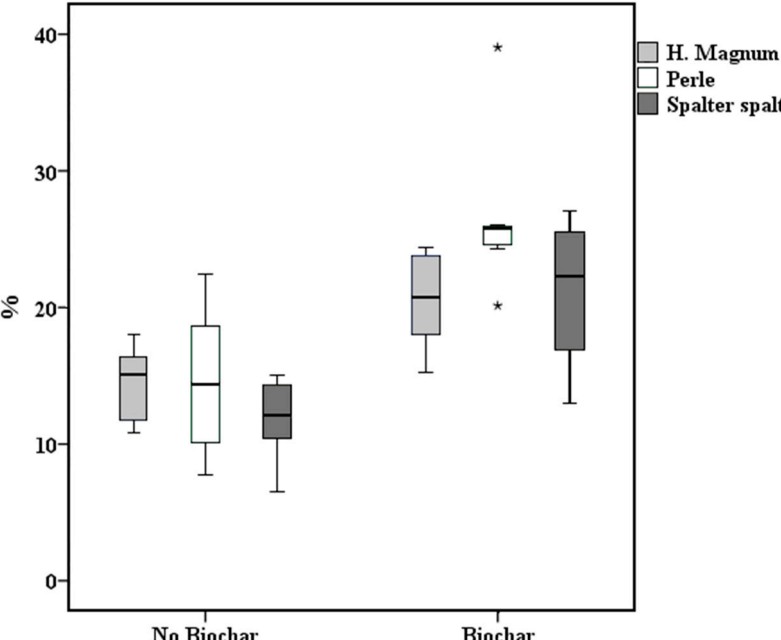

**Figure 3.** Box plots of root percentages of three different hop cultivars (Hallertau Magnum, Perle and Spalter spalt) and two treatments (presence or absence of biochar). Outliers are displayed as filled stars.

### 3.3. Assessment of Hop Plant Response to Biochar Amendment

The evaluation of effect of BSG-derived biochar was carried out using the technique of General Linear Models (GLM). The GLM analysis of roots percentage showed that the two single factors (genotype and treatment) were statistically significant, unlike the two-way interactions between cultivar and treatment. The treatment was the main contributing factor to the percentage of roots ($p < 0.0001$), followed by cultivar ($p = 0.0330$). In fact, as clearly shown in Figure 3, the addition of biochar in the soil produces a greater development of the roots. This phenomenon is even more evident for the Perle cultivar.

Regarding shoot development, specifically the shoot diameter, no variability ascribable to the genotypes was observed for the first three times (Table 1).

**Table 1.** Measurements of diameter of shoots (mean ± standard deviation) at five different times (T1: 30 June, T2: 18 July, T3: 27 July, T4: 3 November, T5: 21 November). Diameters within columns of each cultivar and treatment followed by different letters (a or b) are significantly different.

| Time | Hallertau Magnum | | Perle | | Spalter Spalt | |
|---|---|---|---|---|---|---|
| | No Biochar | Biochar | No Biochar | Biochar | No Biochar | Biochar |
| T1 | 7 ± 2 a | 8 ± 2 a | 8 ± 3 a | 6 ± 1 a | 8 ± 3 a | 5 ± 1 a |
| T2 | 8 ± 3 a | 9 ± 2 a | 9 ± 1 a | 7 ± 1 a | 8 ± 3 a | 7 ± 2 a |
| T3 | 9 ± 2 a | 10 ± 2 a | 10 ± 1 a | 8 ± 1 a | 9 ± 2 a | 8 ± 2 a |
| T4 | 15 ± 3 b | 15 ± 3 b | 11 ± 1 a | 12 ± 2 b | 11 ± 1 a | 12 ± 3 ab |
| T5 | 16 ± 2 b | 16 ± 4 b | 11 ± 1 a | 13 ± 4 b | 11 ± 1 a | 13 ± 3 b |

On the contrary, significant but slight differences resulted among diameters measured before plant dormancy, especially for the Hallertau Magnum cultivar. It is interesting to note that the Perle and Spalter Spalt cultivars tended to be more affected by the presence of biochar during the last phase of plant development, although these differences are not statistically significant. This trend was confirmed by the GLM analysis: only the single genotype and time factors were highly significant ($p < 0.0001$), whereas the single treatment factor, all the two-way interactions and the three-way interaction were not statistically significant.

The GLM analysis on number of leaves at the beginning of May and climbing bines length at harvest highlighted no significant differences between cultivar and treatment and their two-way interaction, as shown in Table 2.

**Table 2.** Measurements of number of leaves (mean ± standard deviation) at the beginning of May, and climbing bines length and number of cones at maturity. Statistical differences between treatments (with biochar or without biochar) for each cultivar and parameter are displayed with different letters (a or b).

| | Hallertau Magnum | | Perle | | Spalter Spalt | |
|---|---|---|---|---|---|---|
| | No Biochar | Biochar | No Biochar | Biochar | No Biochar | Biochar |
| Number of leaves | 20 ± 8 a | 18 ± 6 a | 19 ± 4 a | 15 ± 3 a | 26 ± 14 a | 25 ± 14 a |
| Bines length (cm) | 547 ± 55 a | 570 ± 30 a | 552 ± 27 a | 532 ± 61 a | 555 ± 94 a | 550 ± 75 a |
| Number of cones | 97 ± 17 a | 144 ± 20 b | 84 ± 14 a | 123 ± 18 b | 75 ± 12 a | 106 ± 16 b |

As regards the number of harvested cones, statistical differences ($p < 0.0001$) were observed between treatments for each cultivar: the cones of the plants treated with biochar resulted more numerous compared to those of plants without biochar (Table 2). Instead, the genotype factor and the two-way cultivar-treatment interaction resulted not significant by the GLM analysis.

## 4. Discussion

Despite the high adaptability to different environmental conditions, hop cultivars of European origin are more susceptible to changes in pedo-climatic conditions compared to the American

genotypes [27]. Hops require a large amount of water during the growing seasons to optimize yield and quality. Furthermore, soil has to be nutrient-rich, light, well drained, well supplied with moisture, but free from waterlogging, with an optimal pH range between 6.0 and 6.5., although hops also grow on soils with pH from 4.8 to 8.0 [27].

Our study wanted to verify if biochar can positively affect hop growth. As mentioned above, biochar can improve soil water retention, cation exchange capacity, the content of different nutrients, and reduce soil bulk density and N leaching [7]. Moreover, biochar can enhance the uptake of N and other nutrients by improving the root development and the whole plant physiological status [17]. Then, biochar may reduce plant uptake of potentially toxic elements (such as Pb, Cd and As) from soils, due to the adsorption of metals to the negative charges on the surface of biochar particles [35]. However, the efficacy of biochar application depends on various factors concerning both soil and plant. Understanding these factors is essential for maximizing soil productivity and minimizing potential deleterious environmental effects. For example, the positive effect of biochar amendment is more remarkable in a coarse-textured than in a fine-textured soil, and sandy soils are more responsive than clay-rich soils; this positive effect could be also maximized reducing biochar particle sizes in the range of 0.5–1.0 mm [19]. The biomass feedstock could influence the hydraulic properties of biochar: biochar with high porosity and surface area can improve water adsorption [36].

Previous studies have shown discordant experimental evidences about the supposed beneficial effects of biochar on root growth. Numerous papers reported positive responses on root growth parameters [37–41]. On the contrary, other authors [39,42,43] showed negligible benefits on root growth. These discrepancies could depend on different factors: plant species, growing environment, soil properties, feedstock, biochar properties, pyrolysis conditions [15,21–23]. Furthermore, improved concentrations of dissolved organic C after biochar addition and the adsorption of phenolic compounds by biochar may positively affect root growth [38,43]. Moreover, biochar may stimulate a faster root turnover and changes in root morphology [43]. Although there are few detailed studies on the influence of biochar on the root system, it is generally recognized that plants with a longer root length, a larger surface area and more root tips may be able to get more nutrients and grow better [44–46]. In the present study, the addition of biochar amendment positively affected hop root growth, increasing root length and mass density, as well represented in Figure 2. Moreover, a higher presence of roots could be positively correlated with nutrient uptake, and induce potential benefits in plant growth and development, including aboveground biomass. Benefits and limitations of biochar on plant growth could depend on the types of biochar and the incorporation rates [47–49]. In our study, no statistical differences were observed between number of leaves at the moment of bines elongation and climbing bines length at harvest.

Regarding hop yield, biochar seemed to have positively affected cones production. This production varies greatly during the first years of the plant vegetation. In fact, the hop plant reaches productive and qualitative stability only after the third year. Nevertheless, biochar could have helped the plant when the water requirement was greater. In fact, the summer period, characterized by dry weather conditions, caused long periods of water stress, compensated only by emergency irrigation. Hops require a high water availability for successful plant growth and cones production [27], and changes in soil moisture retention may have been one of the most important factor in explaining positive biochar effects on crop yield. Due to its high porosity and specific surface area, biochar increases soil porosity, and, consequently, the soil hydraulic properties, such as the water storage capacity and the absorption capacity [42]. Furthermore, the induced macro-porosity due to the larger particle size of biochar positively affects soil water permeability [50].

Based on our findings, the application of biochar could be a promising strategy to enhance hop plant growth and cones production. However, further investigation on plant physiological mechanisms and hop cones quality is required to identify optimal levels of biochar application and to better understand the effect of biochar on hop plant growth.

## 5. Conclusions

The present study was conducted to assess the effect of BSG-derived biochar on the hop plant development. The results reinforce the hypothesis that biochar can be a promising and effective soil amendment for hop crops, supplying key nutrients for the plant growth and improving soil properties.

The recovery and reuse of the brewing industry by-products is a good approach of circularity in this industrial sector from the environment protection and waste management standpoint. In fact, the reuse of brewers' spent grain, as feedstock to obtain biochar, will contribute to the adoption of environmentally sustainable practices and a more efficient use of resources, also reducing the use of chemical input. Moreover, it can contribute to decrease the negative economic and environmental impact of the disposal of by-products, decreasing the $CO_2$ production, and increasing the sustainability of the farming sector.

**Author Contributions:** All authors conceptualized and designed the experiments. M.P., V.V. and S.F. collected the data. T.A. performed the statistical analysis and image analysis. T.A. wrote the manuscript in consultation with all other authors. M.P., V.V. and S.F. revised and improved the manuscript. All authors have read and agreed to the published version of the manuscript.

**Funding:** This research received no external funding.

**Acknowledgments:** The valuable technical assistance of Marco Castellani (Department of Agricultural and Forestry Sciences at the University of Tuscia) is gratefully acknowledge. The authors also thank Paola Manzi (CREA Research Centre for Food and Nutrition) for providing the FT-IR instrument.

**Conflicts of Interest:** The authors declare no conflict of interest.

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
