# Peer review of "Evaluation of Spent Grain Biochar Impact on Hop (Humulus lupulus L.) Growth by Multivariate Image Analysis"

_applsci, doi:10.3390/app10020533_

Round 1

Reviewer 1 Report

In my opinion, the manuscript entitled: " Evaluation of spent grain biochar impact on hop 3 (Humulus lupulus L.) growth by multivariate image 4 analysis" it is an interesting contribution however some improvement is necessary. Please see the comments below:

In the summary that biochar supports the growth of cones, but not in the summary in the methodological part of information about cones.

In the introduction part it should be stated how carbon can affect soil microbiota and, as a consequence, the root system. Are there literature reports on this topic.

In the methodological part, please state the original composition (basic characteristics and raw material composition) of wet BSG samples intended for further analysis and how the sample was collected (in what quantity, from which lots).

In the methodical (experimental) part, the sample description should be explained: "with addition of 20% of biochar" - in relation to which 20% was calculated.

In tables 1 and 2, all headings should be capitalized.

At work, it should be taken into account that the variability in crop quality that results from the variety of the plant - in this case hops. As indicated in the work:

Kobus-Cisowska J., Szymanowska D.*, Szczepaniak O.,  Kmiecik D., Przeor M.,  Gramza-MichaÅ‚owska A., Cielecka-Piontek J., Smuga-Kogut M., Szulc P., Composition     and in vitro effects of  cultivars of Humulus lupulus L. hops on cholinesterase activity and microbial  growth", Nutrients 2019, 11, 1377

Author Response

In the summary that biochar supports the growth of cones, but not in the summary in the methodological part of information about cones. Done, see lines 141-144

In the introduction part it should be stated how carbon can affect soil microbiota and, as a consequence, the root system. Are there literature reports on this topic. Done, see lines 73-78

In the methodological part, please state the original composition (basic characteristics and raw material composition) of wet BSG samples intended for further analysis and how the sample was collected (in what quantity, from which lots). Done, see lines 88-90. Regarding the original composition of wet BSG samples we entered only the moisture content. Other measurements can be done on dried samples (see lines 199-200).

In the methodical (experimental) part, the sample description should be explained: "with addition of 20% of biochar" - in relation to which 20% was calculated. Done, see lines 145-150

In tables 1 and 2, all headings should be capitalized. Done

At work, it should be taken into account that the variability in crop quality that results from the variety of the plant - in this case hops. As indicated in the work:

Kobus-Cisowska J., Szymanowska D.*, Szczepaniak O., Kmiecik D., Przeor M., Gramza-Michałowska A., Cielecka-Piontek J., Smuga-Kogut M., Szulc P., Composition and in vitro effects of cultivars of Humulus lupulus L. hops on cholinesterase activity and microbial growth", Nutrients 2019, 11, 1377

As already mentioned in the previous version of the paper, “the hop plant reaches productive and qualitative stability only after the third year”. For this reason, we believe that data on the quality of hop cones may be too variable and unreliable. Therefore, we didn’t carry out quality measurements and we cannot satisfy the referee's request.

Reviewer 2 Report

The experiment is testing beneficial effects of Biochar on hop plant growth and development using only five parameters – root density, leaf number, stem diameter and length and cone number. This is, however, poor description of effects and allow very limited or no conclusions about mechanisms that can be causing the observed positive effects of biochar.

Major flaws of this manuscript are in methodology.

It is unclear how exactly the plants were cultivated. The first period was in in the greenhouse and in 15x14cm pots with artificial soil. Then in May after first measurements is explained that: “plants were planted in the field” Does it mean plants transferred from pots and planted into soil in the hop garden? Or moved and cultivated in pots outside? Both is problematic. Further growth in such small pots till maturity represents serious root growth limitation and, hence will produce results that do not reflect normal physiology of the plant but severe stress. On the other hand – planting plants in the hop garden means a strong reduction of biochar effect unless more biochar was added in the field. In that case, it is likely that more effects were caused by other environmental factors than by biochar.

Why the experimental design was not balanced in the number of replicates in treatment and control is also not explained. Three replicates for control seem also too small number for reliable statistical evaluation.

The novelty of methodology in this paper was indicated by the use of (non-destructive?) evaluation of root development, namely root density using image analysis. Here again, the most important part of the description is missing. Namely, how the pictures for analysis were precisely acquired. I assume that plants were extracted from pots before measurement but this is not stated in the text. Was the whole system spread and photographed or just one side of soil facing the pot wall? Why this was not done at the end of the experiment as well?

Authors also did not take a chance to extract more useful and detailed information from image analysis. This could be for example more info about various root diameters and their contribution to total root length or information about root branching frequency. This information may be much more useful for the explanation of biochar effects.

It is also not clear why authors did not use for evaluation of effects also plant biomass and leaf area as key parameters of plant physiological productivity.

Presentation of results is also problematic and incomplete – authors do not give clear information about what is presented – presumably mean value but what is the measure of variability? What was the test used for testing significance? What was the level of significance used? In tables is the significance valid only for biochar treatment or also across cultivars used? Or even measurement times in Table 1? Completely unclear.

In conclusion, sophisticated methods of image analysis and data processing including statistical evaluation are based on data that were collected by a very unclear way. Unless this part is clarified it is very problematic to draw any reliable conclusion. Discussion about the putative positive effects of biochar on hop plants here can only be based on speculation without any hint of a mechanistic explanation of observed positive effects. I think this is too little for a scientific paper.

Author Response

Reviewer 3

The experiment is testing beneficial effects of Biochar on hop plant growth and development using only five parameters – root density, leaf number, stem diameter and length and cone number. This is, however, poor description of effects and allow very limited or no conclusions about mechanisms that can be causing the observed positive effects of biochar.

Major flaws of this manuscript are in methodology.

It is unclear how exactly the plants were cultivated. The first period was in in the greenhouse and in 15x14cm pots with artificial soil. Then in May after first measurements is explained that: “plants were planted in the field” Does it mean plants transferred from pots and planted into soil in the hop garden? Or moved and cultivated in pots outside? Both is problematic. Further growth in such small pots till maturity represents serious root growth limitation and, hence will produce results that do not reflect normal physiology of the plant but severe stress. On the other hand – planting plants in the hop garden means a strong reduction of biochar effect unless more biochar was added in the field. In that case, it is likely that more effects were caused by other environmental factors than by biochar.

We clarified as the experiments was carried out. See lines 144-146

Why the experimental design was not balanced in the number of replicates in treatment and control is also not explained. Three replicates for control seem also too small number for reliable statistical evaluation.

We clarified it. See lines 144-146

The novelty of methodology in this paper was indicated by the use of (non-destructive?) evaluation of root development, namely root density using image analysis. Here again, the most important part of the description is missing. Namely, how the pictures for analysis were precisely acquired. I assume that plants were extracted from pots before measurement but this is not stated in the text. Was the whole system spread and photographed or just one side of soil facing the pot wall? Why this was not done at the end of the experiment as well?

We clarified it. See lines 156-163

Authors also did not take a chance to extract more useful and detailed information from image analysis. This could be for example more info about various root diameters and their contribution to total root length or information about root branching frequency. This information may be much more useful for the explanation of biochar effects.

According to us, it is necessary remove the soil to evaluate the required aspects (root diameters and their contribution to total root length or information about root branching frequency). But this is contrary to the aims of our study, namely that of rapidly and non-destructively assessing the differences in the density of the root system

It is also not clear why authors did not use for evaluation of effects also plant biomass and leaf area as key parameters of plant physiological productivity.

As already mentioned in the paper, “the hop plant reaches productive and qualitative stability only after the third year”. For this reason, we believe that the required assessments requested are not significant. Surely, it is a suggestion that can be collected for studies after the third year of the plant's life.

Presentation of results is also problematic and incomplete – authors do not give clear information about what is presented – presumably mean value but what is the measure of variability? What was the test used for testing significance? What was the level of significance used? In tables is the significance valid only for biochar treatment or also across cultivars used? Or even measurement times in Table 1? Completely unclear.

We clarified them. See lines 192-193; 257, 271

In conclusion, sophisticated methods of image analysis and data processing including statistical evaluation are based on data that were collected by a very unclear way. Unless this part is clarified it is very problematic to draw any reliable conclusion. Discussion about the putative positive effects of biochar on hop plants here can only be based on speculation without any hint of a mechanistic explanation of observed positive effects. I think this is too little for a scientific paper.

The aim of this study is to verify if biochar could positively affect the hop plant and if the image analysis coul help to assess the changes in plant growth. These objectives have been achieved. However, the reviewer suggested new assessments that can be made only after the third year of the hop plant's life. For other types of plants, that reach maturity already in the first year of development, the suggestions would have already been included in the work. Furthermore, image analysis could also be used successfully to evaluate leaf development and production. In the case of hops, the plant rises above 7 meters and, therefore, it is difficult to have an adequate resolution of the image. This study can be a useful tool to replicate and deepen for other crops

Reviewer 3 Report

Dear authors,

This manuscript constitutes an interesting and well performed study concerning the assessment of the potential ability of BSG-derived biochar as soil amendment and its influence on hop plant growth. Indeed, this manuscript gives interesting insights about the subject. The overall idea is interesting and the methodology is very well described. The manuscript, in general, is really clear and well structured. I think that the interpretation and discussion of the results could be enriched and improved. 

Author Response

This manuscript constitutes an interesting and well performed study concerning the assessment of the potential ability of BSG-derived biochar as soil amendment and its influence on hop plant growth. Indeed, this manuscript gives interesting insights about the subject. The overall idea is interesting and the methodology is very well described. The manuscript, in general, is really clear and well structured. I think that the interpretation and discussion of the results could be enriched and improved.

The discussion and references have been enriched and improved; see lines 281-335.

Round 2

Reviewer 2 Report

The manuscript did not substantially change after revision. It only shows that biochar MAY have a positive effect on root growth of young hop plants.

I think that both the use of image analysis and use of biochar can bring a piece of useful information but to show benefits of both a different approach is necessary.

To show the usefulness of the image analysis method this method must be verified first. Now authors assume that the area of roots in contact with the wall of the pot is closely correlated with the root density in the whole pot but show no data to confirm this. So, here a separate experiment is needed to verify the presented “novel” method by using classical destructive methods (showing differences in root length, diameter, biomass, SRL etc.) and how these parameters correlate by those obtained by image analysis. This can also show more in detail what beneficial effects on root growth (as the only positive effect in this work) may be caused by biochar. Such info may be useful e.g. for commercial producers of hop seedlings when biochar potentially helps to better establish seedling in the field. I can also imagine comparison of effects of biochar from different sources using fast, non-destructive image analysis of root system.

I do not comment on minor problems of this MS such as the way of presentation (legends to tables and figures are not self-explanatory and some important info is still missing e.g. measurement units, the meaning of symbols) or some language problems.

Without more trustful and detailed data about the effect of biochar on root growth is the whole discussion in this manuscript only speculation. It could be used as a review about potential positive effects of biochar but it does not bring useful information in the context of presented results. So, I can see the potential for the usefulness of this work when appended by more experiments but I do not recommend publication in the present form.

Author Response

The authors reiterate that the aim of this study was to evaluate the effect of biochar on hop plants only with non-destructive techniques.
Further considerations may be the subject of a new study, in which some of the aspects that the reviewer emphasizes may be considered.